# Preparation of a Flame-Retardant Curing Agent Based on Phytic Acid–Melamine Ion Crosslinking and Its Application in Wood Coatings

**DOI:** 10.3390/polym16111557

**Published:** 2024-05-31

**Authors:** An Wei, Shunxiang Wang, Yongjin Zou, Cuili Xiang, Fen Xu, Lixian Sun

**Affiliations:** 1College of Materials Science and Engineering, Guilin University of Electronic Technology, Guilin 541004, China; 2Nanning Guidian Electronic Technology Research Institute Co., Ltd., Nanning 530000, China

**Keywords:** phytic acid, urea–formaldehyde resin, flame-retardant coating, fire-retardant wood

## Abstract

To broaden the applications of wood, it is necessary to prepare flame-retardant coatings that can protect wood substrates during combustion. In this study, a bio-based, intumescent, flame-retardant phytic acid–melamine polyelectrolyte (PM) was prepared using phosphorus-rich biomass phytic acid and nitrogen-rich melamine as raw materials through an ion crosslinking reaction. Subsequently, a series of bio-based, flame-retardant wood coatings were prepared by optimizing the structure of urea–formaldehyde resin with the addition of melamine, sodium lignosulfonate, and PM as a flame-retardant curing agent. Woods coated with PM-containing coatings displayed significantly improved flame-retardant performances in comparison to uncoated woods. For PM-cured woods, the measured values of total heat release and total smoke production were 91.51% and 57.80% lower, respectively, compared with those of uncoated wood. Furthermore, the fire growth index decreased by 97.32%, indicating a lower fire hazard. This increase in flame retardancy and smoke suppression performance is due to the dense expanded carbon layer formed during the combustion of the coating, which isolates oxygen and heat. In addition, the mechanical properties of the flame-retardant coatings cured with PM are similar to those cured with a commercial curing agent, NH_4_Cl. In addition, the prepared flame-retardant coating can also stain the wood. This study proves the excellent flame-retarding and curing effect of ammonium phytate in urea–formaldehyde resin coatings and provides a new approach for the application of bio-based flame retardants in wood coatings.

## 1. Introduction

Wood is a resource-rich, green, renewable, and highly processable material. It is widely used in fields such as construction and furniture production and is closely linked to people’s lives [1,2,3]. However, wood itself is flammable, and destructive fires often occur due to wood burning [4]. Therefore, it is necessary to carry out modifications of wood in order to improve its flame-retardant performance and reduce its fire hazard [5,6]. Inorganic flame retardants alone do not readily adhere to the surface of wood; as such, using a flame-retardant adhesive coating is currently the most used method for flame retardancy [7,8]. Flame-retardant coatings can improve not only the flame-retardant performance of wood but also the physical and chemical properties of the wood surface [9].

Acrylic [10], epoxy [11], and amino resins [12] are commonly used in wood coatings. These resins have high adhesion and good compatibility with wood surfaces. However, these coatings also have a certain degree of inherent flammability, and flame-retardant modifications are required to prepare them for use as flame-retardant coatings [13,14]. Halogen-based flame retardants have traditionally been the flame retardants of choice; however, they generate harmful substance during combustion and have been gradually banned from use [15]. Intumescent flame retardants composed of phosphorus and nitrogen have begun to replace halogen flame retardants, owing to their low toxicity, strong smoke suppression ability, and high flame-retardant efficiency [16,17,18]. Intumescent flame retardants generate phosphoric acid derivatives during the combustion process, which promote the dehydration and densification of the wood substrate while simultaneously forming a dense, protective carbon layer to protect it [19]. Intumescent flame retardants also release non-combustible gases to dilute oxygen and expand the carbon layer, reducing the degree of combustion [20].

Based on the demand for environmental protection, biodegradable biomass materials from a wide range of sources have gained increasing attention from researchers [21,22,23]. Owing to their unique structures, many biomass materials have been used to prepare bio-based flame retardants, including phytic acid (PA) [24,25], chitosan [26,27], lignin [28,29], cashew phenols [30,31], and eugenol [32]. These bio-based flame retardants can be converted into intumescent flame retardants via the addition of phosphorus and nitrogen [33,34]. With a phosphorus content of up to 27.3%, PA is a suitable acid source for preparing intumescent flame retardants. At the same time, it also has a very strong chelating ability, making it easy to combine with positively charged cations or metal ions to form polyelectrolytes and metal salts [35,36].

Urea–formaldehyde (UF) resin, a type of amino resin, is widely used in both wood adhesives and wood coatings [37,38]. This resin is rich in nitrogen and can undergo modifications that enhance its properties. For instance, modifying UF with phosphorus would result in an intumescent flame-retardant wood coating containing both phosphorus and nitrogen. Urea–formaldehyde resins can be further improved via modification with melamine (MEL). MEL provides nitrogen and can also improve the water resistance of flame-retardant systems. Sodium lignosulfonate is another additive used to provide a carbon source for intumescent flame-retardant systems and improve the storage stability of urea–formaldehyde resins [39]. Finally, curing agents are important additives in flame-retardant coatings. Commercial curing agents such as ammonium chloride—commonly used in urea–formaldehyde resins—contain halogens. Similar to halogen-based flame retardants, these halogen-based curing agents will produce harmful substances during the combustion process. To avoid the potential hazards of halogen combustion, a new halogen-free curing agent is required to replace ammonium chloride.

In this work, we utilized ion crosslinking between phytic acid and MEL to prepare a bio-based, flame-retardant curing agent, phytic acid–melamine polyelectrolyte (PM). This curing agent was then incorporated into MEL and sodium-lignosulfonate-modified urea–formaldehyde resins to prepare a series of flame-retardant coatings. The structural, thermal, physical, and flame-retardant properties of the coatings were carefully studied and compared with an NH_4_Cl-cured control coating to elucidate the effects of PM. Compared with NH_4_Cl-cured coatings, PM-cured coatings not only have similar physical and mechanical properties but also significantly improve flame retardancy. In addition, due to the presence of sodium lignosulfonate, the prepared flame-retardant coating can also dye the material and improve its aesthetic effect. The use of PM has demonstrated the excellent flame-retardant and curing effect of bio-based ammonium phytate in urea–formaldehyde resin coatings. Thus, this study provides a new approach for the application of bio-based components in flame-retardant coatings for wood.

## 2. Materials and Methods

### 2.1. Materials

Phytic acid (PA, 70 wt.% in water, CAS number: 83-86-3), formaldehyde solution (37 wt.% in water, CAS number: 50-00-0), urea (CAS number: 57-13-6), melamine (MEL, CAS number: 108-78-1), sodium lignosulfonate (CAS number: 8061-51-6), ammonium chloride (CAS number: 12125-02-9), and dioctyl phthalate (DOP, CAS number: 117-81-7) were obtained from Aladdin Scientific Corp. (Shanghai, China). Sodium hydroxide solution (20 wt.% in water) and acetic acid solution (20 wt.% in water) were prepared in house.

### 2.2. Preparation of Samples

#### 2.2.1. Preparation of PM

Over the course of 30 min, MEL (25.3 g) was dissolved in deionized (DI) water (400 mL) at 85 °C at a stirring speed of 600 rpm. Concurrently, PA (70 wt.% solution in water, 24.5 mL) was combined with DI water (50 mL). The prepared PA solution was slowly added to the prepared MEL solution, allowing MEL and PA to fully react. After reaction completion, the obtained product was washed using DI water until the pH of the washing solution was within 4.0–5.0. The filtered residue was vacuum-dried at 80 °C for 24 h and then ground into a powder to obtain PM.

#### 2.2.2. Preparation of SUF Coatings and Their Application on Wood

Formaldehyde solution (37 wt.% in water, 100 g) was added to a flask. The pH of the solution was adjusted using sodium hydroxide solution (20 wt.% in water) to 8.0–8.5. The solution was mechanically stirred and heated to 90 °C. Urea (37 g) was added at 90 °C, followed by MEL (0.57 g) and sodium lignosulfonate (5.7 g). This temperature was maintained with continuous stirring for 30 min. Subsequently, the pH was adjusted to 4.5–5.0 using acetic acid solution (20 wt.% in water). A second batch of urea (12.3 g) was added, and the solution was stirred continuously for 10 min. Subsequently, the pH was adjusted to 7.5–8.0 using sodium hydroxide solution (20 wt.% in water), and a third batch of urea (7.6 g) was added. The temperature was reduced to 70 °C and stirred for 30 min. Cool the solution to room temperature (25 °C) to obtain a modified UF (MUF) resin. Subsequently, DOP (1 g) was added to MUF (100 g), followed by a curing agent. The mixture was stirred for 2 h at a speed of 700 rpm to obtain the sodium-lignosulfonate-modified urea–formaldehyde resin (SUF) flame-retardant coating. A corresponding flame-retardant wood sample (SUFW) was obtained by applying a 0.3 mm-thick layer of the SUF coating to the wood. The formulation of each SUF coating is shown in Table 1.

### 2.3. Sample Characterization

#### 2.3.1. Structural Characterization

Fourier transform infrared (FTIR) spectroscopy was performed using a TENSOR27 infrared spectrometer (BRUKER, Saarbrücken, Germany). The microstructures and element distribution of residual carbon layers were observed by scanning electron microscopy (SEM) and energy dispersive spectrometry (EDS) with a Quanta 450 FEG field-emission scanning electron microscope (FEI, Hillsboro, OR, USA). The degree of graphitization of coating residue was analyzed via Raman spectroscopy using a HORIBA LabRAM HR Evolution Raman spectrometer (HORIBA Scientific, Paris, France) at a laser excitation wavelength of 532 nm.

#### 2.3.2. Thermal Analysis

The curing behavior of the materials was analyzed via differential scanning calorimetry (DSC) with a Star449 F3 thermal analyzer (NETZSCH Group, Saarbrücken, Germany) under nitrogen atmosphere. Samples were analyzed from 40 to 180 °C with a heating rate of 10 °C/min. An Is-50 thermogravimetric analyzer (Thermo Fisher Scientific, Waltham, MA, USA) was used for thermogravimetric (TG) analysis. Samples were analyzed from 50 to 800 °C with a heating rate of 20 °C/min and a nitrogen flow rate of 100 mL/min.

#### 2.3.3. Flame Retardance Tests

Underwriters Laboratories (UL) 94 vertical combustion tests were performed using a CZF-3 vertical combustion test instrument (Jiangning Analytical Instrument, Nanjing, China) in accordance with ASTM International Standard D3801 [40]. Limiting oxygen index (LOI) tests were performed according to ASTM International Standard D2863 [41] using an HC-2 oxygen index meter (Jiangning Analytical Instrument, Nanjing, China). Cone calorimetry tests were conducted using a TTech-GBT161172 cone calorimeter (Testech Technology, Suzhou, China) according to ASTM International E1354 [42]/ISO 5660 standards [43].

#### 2.3.4. Analysis of Physical Properties

The hardness of each coating was characterized using a QHQ-A pencil hardness tester (Huaguo Precision Technology, Dongguan, China) in compliance with ASTM International Standard D3363-2005 [44]. The adhesion of each coating was characterized using a QHF hundred grid knife (Huaguo Precision Technology, Dongguan, China) in compliance with ASTM International Standard D3359-09 [45]. The glossiness of each coating was characterized using an HST-60A glossmeter (Huiste, Shenzhen, China) in compliance with ASTM International Standard D523 [46]. The water resistances of the coatings were determined according to ASTM International Standard D870-02 [47]. The water resistances of SUFW were estimated by monitoring for color changes and foaming after soaking at 30 °C for 48 h.

## 3. Results and Discussion

### 3.1. FTIR Analysis of the PM Structure

The FTIR spectra of PA, MEL, and PM are shown in Figure 1. We observed absorption bands in MEL at 3470 cm^−1^ and 3420 cm^−1^, which were attributed to the –NH_2_ functional group; these bands were not observed in the PM spectrum. A broad peak appearing at 3120 cm^−1^ in both the MEL and PM spectra indicated the presence of the –NH_3_^+^ functional group [48]. The characteristic peaks of the triazine ring at 1648 cm^−1^, 1552 cm^−1^, and 1431 cm^−1^ in MEL were shifted in the PM spectrum, possibly due to the deformation of the triazine ring in PM [49]. Finally, absorption bands characteristic of PO_4_^3−^, P–O–C, and P=O motifs were observed at 522 cm^−1^, 983 cm^−1^, and 1177 cm^−1^, respectively, in PM. Peaks representing the same functionalities were also clearly visible in the PA starting material. Together, these results indicated that PA and MEL successfully reacted to produce PM.

### 3.2. Thermal Performance of Flame-Retardant SUF Coating

DSC can be used to evaluate the curing processes of materials. We performed a DSC analysis for each of the prepared SUF coatings; the results are shown Figure 2. All coatings exhibited obvious heat release peaks. SUF-0, cured with NH_4_Cl, and SUF-1, cured with an equivalent amount of PM, displayed similar DSC curves with comparable peak curing temperatures. This indicated that the curing effect of PM was similar to that of NH_4_Cl. Comparing the curves of SUF-1, SUF-1.5, and SUF-2, we observed a gradual decrease in the peak curing temperature of the coating as the amount of PM in the formulation increased. This indicates that increasing the amount of PM in the formulation accelerates the curing of the coating. Altogether, the DSC results indicate that PM can effectively cure the resin.

The TG analysis results of the flame-retardant SUF coatings are presented in Figure 3 and Table 2. The thermal degradation behavior did not vary greatly amongst the prepared coatings. We determined that the initial decomposition temperature (*T*_d5%_) of SUF-1 was 16.5 °C higher than that of SUF-0. SUF-1 also exhibited a higher maximum decomposition rate temperature (*T*_peak_) of 305.2 °C in comparison to SUF-0, which exhibited a *T*_peak_ of 275.1 °C. These results indicated that the use of PM in place of NH_4_Cl increased the thermal stability of the coating. SUF-1.5 and SUF-2, which contained higher amounts of PM than SUF-1, both exhibited decreased *T*_d5%_ values; in comparison to that of SUF-1, the *T*_peak_ values of SUF-1.5 and SUF-2 were slightly higher and lower, respectively. However, the magnitudes of these decreases were not significant, and both the *T*_d5%_ and *T*_peak_ values of SUF-2 were still higher than those of SUF-0. Additionally, the residual mass of SUF-1 at 800 °C (*R*_800_) was 229% that of SUF-0 at the same temperature. As the amount of PM in the formulation increased, the *R*_800_ value also increased; SUF-2 exhibited an *R*_800_ value of 37.4 wt.%. We attribute this finding to the presence of PO_4_^3−^ groups in PM, which dehydrates the matrix into carbon during high-temperature decomposition, forming a protective carbon layer that prevents further decomposition.

### 3.3. Flame-Retardant Performance of SUFW

The flame-retardant performances of the SUFW were characterized through UL-94 and LOI testing. The results of these tests are shown in Figure 4 and Table 3. The use of coatings turned the wood brown. Pure wood burned completely in the vertical combustion test and underwent significant fragmentation; while some charred wood fell from the specimen, the cotton was not ignited. The SUFW-0 sample exhibited a decreased extent of combustion in comparison to that of the pure wood. SUF-1 outperformed SUF-0 in this context, and the extent of combustion continued to decrease as the PM content in the coating increased. While the wood coated with SUF-0 only reached a V-1 level in the vertical combustion test and achieved an LOI of 27.2%, all three wood samples cured with PM (SUF-1, SUF-2, and SUF-3) achieved the highest vertical combustion test designation of V-0. Furthermore, the LOI of SUFW-2 reached 32.1%, indicating that increasing the PM content further enhanced the flame-retardant performance of the wood.

In addition, the SUFW-2 sample prepared in this work was simply compared with flame-retardant-treated wood or other materials such as polylactic acid [50] in other studies. As shown in Table 4, the flame retardancy of SUFW-2 is comparable or even better than that of other materials treated with flame retardants. The flame-retardant materials in other studies, such as TPLA/PA-HDA_10_, contain up to 10% flame retardant, while the amount of PM used in this work is only 2%. Therefore, the flame-retardant curing agent used in this work has a low amount of addition and better flame-retardant performance.

Cone calorimetry is a method commonly used to simulate real combustion conditions. SUFW samples were analyzed via this method (show in Figure 5 and Table 5). Compared with pure wood, all wood samples coated with flame-retardant coating exhibited an increase in time to ignition (TTI) and a significant decrease in peak heat release rate (PHRR) and total heat release (THR) (Figure 5a,b). Relative to those of pure wood, the THR and PHRR values of SUFW-0 were decreased by 71.96% and 73.4%, respectively. The PM-containing SUFW-1 exhibited values of THR and PHRR that were decreased by 87.82% and 82.45%, respectively, compared to those of pure wood; additionally, the TTI of SUFW-1 was 5 s longer than that of pure wood. It was found that increasing the amount of PM in the sample coating further decreased the THR and PHRR values. This indicates that PM effectively inhibits the heat release of wood when incorporated into the sample coating. In addition, the total smoke production (TSP) also decreased with increasing amounts of PM in the sample coating (Figure 5c). PM-containing SUFW samples can likely form a thicker expanded carbon layer, which hinders smoke emissions. The fire growth index (FGI) of the sample decreased by 97.32% from 2.24 for pure wood to 0.06 for SUFW-2, with the PM-containing coatings performing the best. This indicates that the fire hazard of wood is significantly decreased after coating with flame-retardant coatings. This was further reflected in the degree of mass loss of the analyzed samples during cone calorimetry (Figure 5d). The residual amount of pure wood was less than 30% after cone calorimetry, while wood samples with flame-retardant coatings retained 70% or more of their masses. This demonstrates that all the studied flame-retardant coatings protected the base wood during the combustion process, resulting in better mass retention of the sample.

Images of the sample before and after the cone calorimetry test and the height of the expanded carbon layer formed after the test are shown in Figure 6. Pure wood burned completely during testing, resulting in residual broken charcoal. All SUF-coated wood formed an expanded carbon layer to protect the substrate. We observed that the PM-coated wood samples formed taller expanded carbon layers than the NH_4_Cl-coated wood sample; the height of the expanded carbon layer increased with an increasing amount of PM in the coating. The height of the expanded carbon layer is indicative of the protective ability of the wood coating.

### 3.4. Investigating the Flame-Retardant Mechanism

To better understand the flame-retardant mechanism, each wood sample was analyzed via SEM and EDS after cone calorimetry. The results are presented in Figure 7. The residual carbon of the pure wood sample after combustion was broken and uneven, while the residual carbon of SUFW-0 possessed a relatively dense morphology; however, it still exhibited many large pores and wrinkles caused by gas flushing. The residual char morphology of SUFW-2 was denser and more continuous; almost no pores were observed, indicating that the quality of the residual char layer formed by SUFW-2 was more robust and that the layer was better able to protect the wood substrate. We attribute this result to the presence of phosphorus groups in the residual char and speculate that phosphate derivatives promote the crosslinking of the char layer, increasing its density.

The ratio of the intensities of the D-band (*I*_D_) and the G-band (*I*_G_) in Raman spectroscopy can be used to characterize the degree of graphitization of a material. The smaller the ratio (*I*_D_/*I*_G_), the higher the degree of graphitization of the material. The Raman spectroscopy results for this study are shown in Figure 8. The value of *I*_D_/*I*_G_ for the pure wood residue was 2.64, indicating a relatively low degree of graphitization. The *I*_D_/*I*_G_ value for the residual carbon of SUF-0 was determined to be 2.30. The *I*_D_/*I*_G_ value of the residual carbon from SUFW-2, however, was 1.98; this indicates a significant increase in the degree of graphitization compared to the values for pure wood and SUFW-0. These results were in good agreement with those collected by SEM.

The flame-retardant coating is crucial for the protection of the wooden substrate during combustion. The phosphate groups are key to this process; they promote the dehydration of the carbon layer during combustion, forming a dense carbon layer that isolates heat and oxygen. This isolation in turn limits further combustion of the substrate. Owing to its rich carbon and nitrogen content, the coating also produces non-combustible gases during this process, such as NH_3_ and CO_2_. These non-combustible gases dilute oxygen in the gas phase and expand the carbon layer, providing better protection for the wood substrate. Our results are therefore representative of a typical expansion flame-retardant mechanism (Figure 9) [51].

### 3.5. Physical Properties of the Wood Coatings

The physical properties of the coatings are very important for practical application. The hardness, adhesion, glossiness, and water resistance of the SUF-coated woods were tested; the results are listed in Table 6. SUFW-0 exhibited a measured pencil hardness of 9H, while SUFW-1, SUFW-1.5, and SUFW-2 achieved a slightly lower hardness level of 8H. However, this is still a higher hardness than that of most coatings. All coatings achieved an adhesion level of 5B. The reason for the high adhesion is that hydrogen bonds may be formed between the paint and the wood surface, and the existence of hydrogen bonds can make the paint and the wood surface have a high affinity for each other [52]. We also determined that the glossiness of the samples decreased with the increasing PM content of the coating. However, the glossiness of all four coatings exceeded 30. Finally, the water resistance of each coating was determined by monitoring for changes in the coating after the sample was immersed in water for 48 h. No significant changes in the coatings of SUFW-0 or SUFW-1 were observed. However, the coatings of SUFW-1.5 and SUFW-2 showed slight detachment. We speculate that this is due to the increased PM content of these coatings, which leads to an increase in the content of hydrophilic groups in the coating. Altogether, the results of the physical performance tests show that the physical properties of PM- and NH_4_Cl-cured coatings are similar. As such, PM could be a promising replacement for NH_4_Cl, capable of meeting the typical requirements of wood coatings.

## 4. Conclusions

This work is based on the ion crosslinking effect of PA and MEL to prepare a flame-retardant curing agent, PM. Flame-retardant coatings of urea–formaldehyde resin modified with MEL and sodium lignosulfonate were obtained using PM as a flame-retardant curing agent. SUFW-2 coated with the flame-retardant coating not only passed UL-94 V-0 level testing but also had an LOI of 32.1%. SUFW-2 exhibited a 95.44% and 91.51% lower PHRR and THR, respectively, in comparison to pure wood, and a 57.80% lower TSP, demonstrating good flame retardancy and smoke suppression performance. The mechanisms of the coatings are mainly due to the formation of an expanded carbon layer during the combustion process, which protects the wooden substrate and reduces smoke generation. The physical properties of the coatings cured with PM are similar to those of the coatings cured with NH_4_Cl, a commercial curing agent. However, the increase in PM content will reduce the water resistance performance of the coating, which requires further improvement. Compared with existing work, the flame-retardant curing agent PM prepared in this work has the advantages of simple preparation, low usage, and excellent effect on improving flame-retardant performance. The work has proven that ammonium phytate can be effectively applied to the flame-retardant curing of urea–formaldehyde resin coatings. In addition, due to the presence of sodium lignosulfonate, the prepared coatings can also dye the materials, which helps to improve their aesthetics. Taken together, these findings demonstrate a new approach for the application of bio-based flame retardants in flame-retardant coatings.

## Figures and Tables

**Figure 1 polymers-16-01557-f001:**
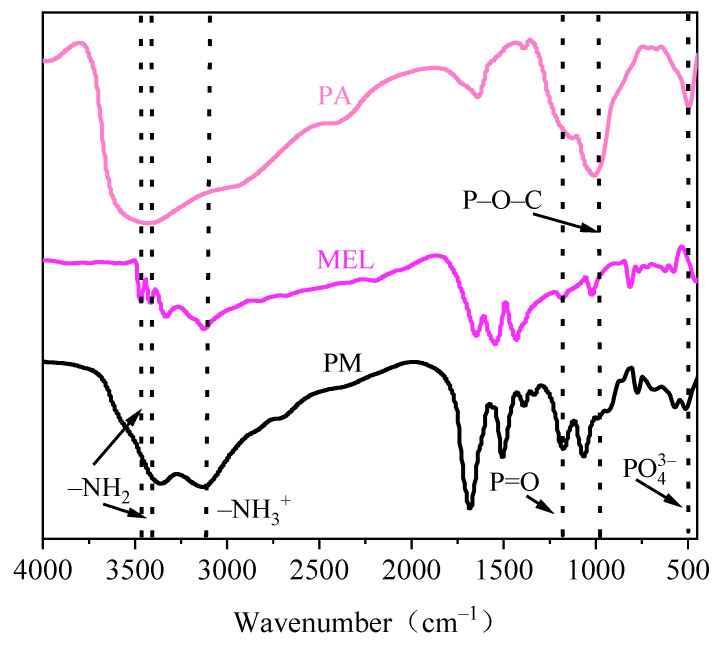
FTIR images of phytic acid (PA), melamine (MEL), and PM.

**Figure 2 polymers-16-01557-f002:**
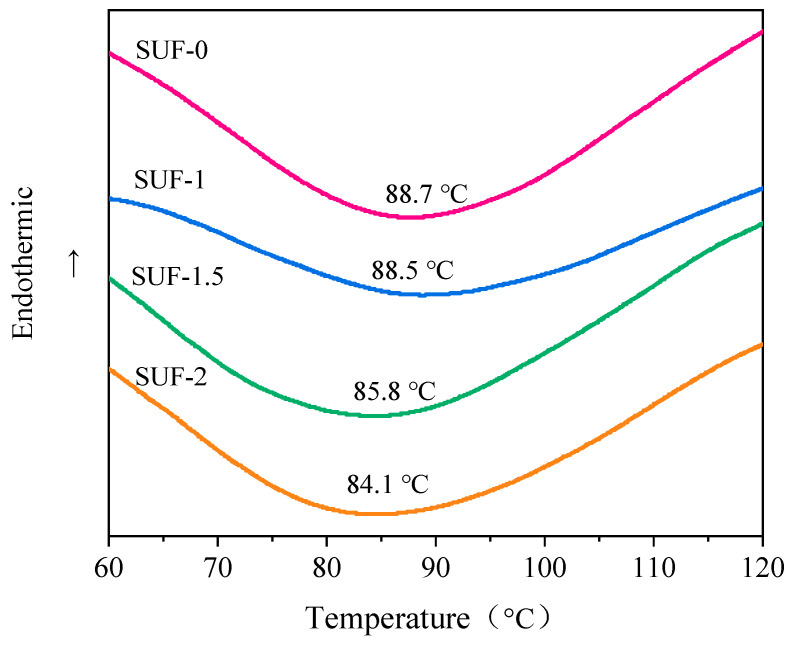
DSC curves of the flame-retardant SUF coatings.

**Figure 3 polymers-16-01557-f003:**
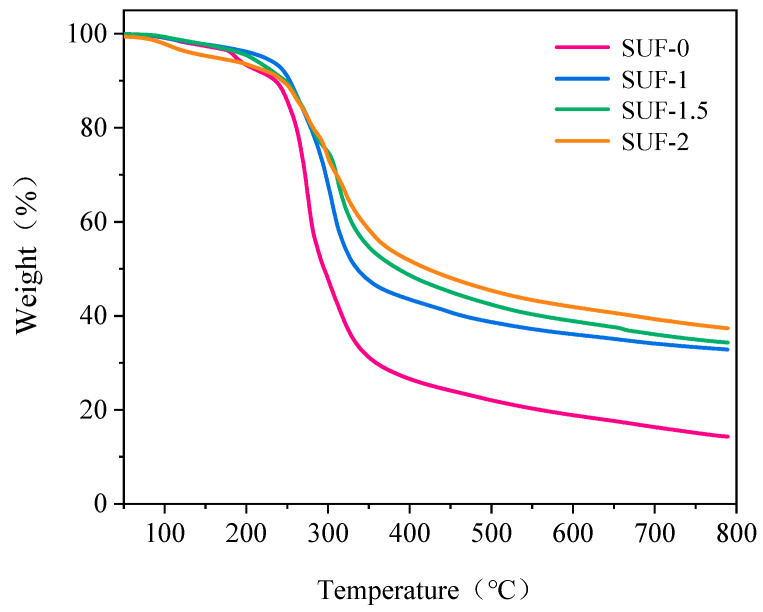
Thermogravimetric (TG) analysis curves of the SUF coatings.

**Figure 4 polymers-16-01557-f004:**
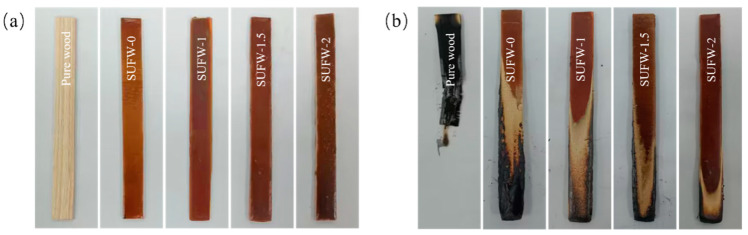
Results of the vertical combustion experiment for pure wood and SUF-coated woods. (**a**) Wood samples before UL-94 testing; (**b**) Wood samples after UL-94 testing.

**Figure 5 polymers-16-01557-f005:**
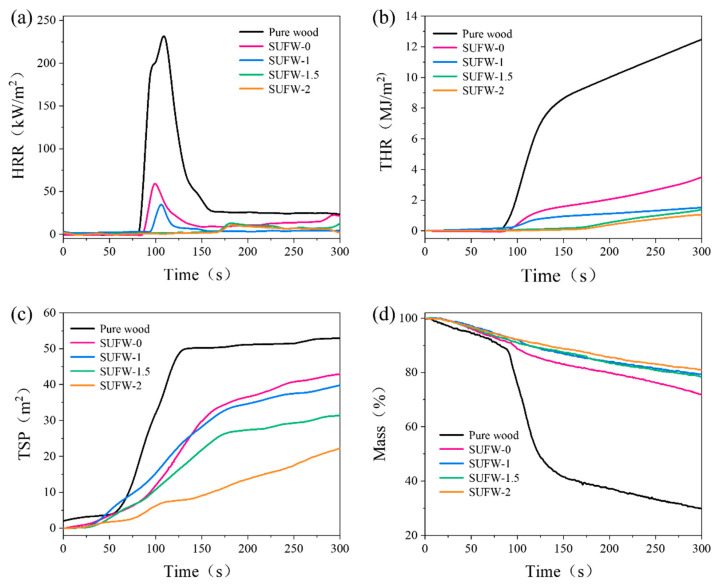
Cone calorimetry results for pure and SUF-coated wood samples. (**a**) HRR plot; (**b**) THR plot; (**c**) TSP plot; (**d**) mass loss plot.

**Figure 6 polymers-16-01557-f006:**
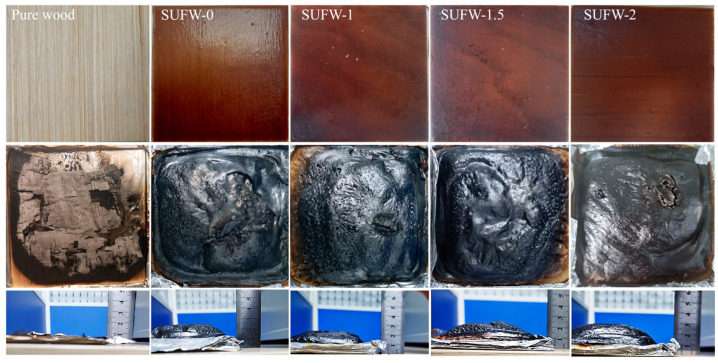
Images of wood samples before (**top row**) and after (**middle row**) cone calorimetry. (**Bottom row**): height map of the formed expanded carbon layer.

**Figure 7 polymers-16-01557-f007:**
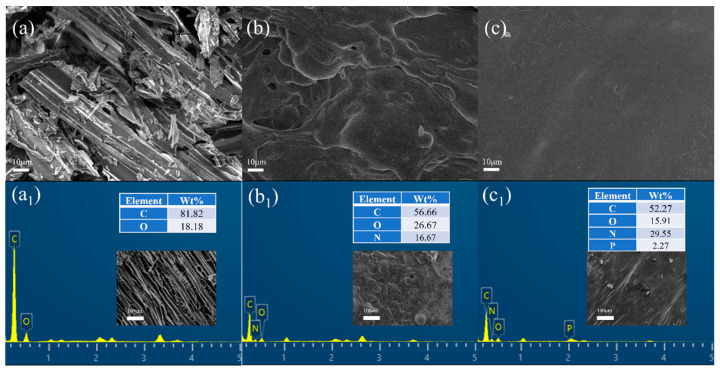
SEM (**top row**) and EDS (**bottom row**) images of residual carbon layers of the wood samples after cone calorimetry testing. (**a**,**a_1_**) Pure wood; (**b**,**b_1_**) SUFW-0; (**c**,**c_1_**) SUFW-2.

**Figure 8 polymers-16-01557-f008:**
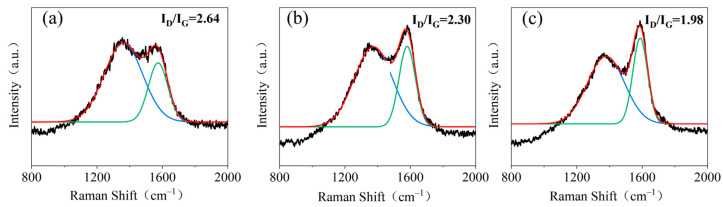
Raman spectra of residual char from pure and SUF-coated woods. (**a**) Pure wood; (**b**) SUFW-0; (**c**) SUFW-2.

**Figure 9 polymers-16-01557-f009:**
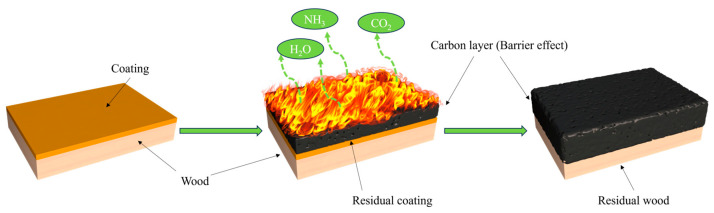
Burning process of the SUF-coated woods.

**Table 1 polymers-16-01557-t001:** Formulations of sodium-lignosulfonate-modified urea–formaldehyde resin (SUF) flame-retardant coatings. The coatings were produced from combinations of modified urea–formaldehyde (MUF) resin, dioctyl phthalate (DOP), NH_4_Cl, and phytic acid–melamine polyelectrolyte (PM).

Coating	MUF (g)	DOP (g)	NH_4_Cl (g)	PM (g)
SUF-0	100	1	1	0
SUF-1	100	1	0	1
SUF-1.5	100	1	0	1.5
SUF-2	100	1	0	2

**Table 2 polymers-16-01557-t002:** Initial decomposition temperatures (*T*_d5%_), maximum decomposition rate temperatures (*T*_peak_), and residual masses at 800 °C (*R*_800_) for the SUF coatings as measured by TG analysis.

Sample	*T*_d5%_ (°C)	*T*_peak_ (°C)	*R*_800_ (wt.%)
SUF-0	235.6	275.1	14.3
SUF-1	253.1	305.2	32.8
SUF-1.5	248.3	312.2	34.5
SUF-2	244.9	297.7	37.4

**Table 3 polymers-16-01557-t003:** Results of the UL-94 and limiting oxygen index (LOI) tests for pure and SUF-treated wood; the first and second burning times are denoted *t*_1_ and *t*_2_, respectively.

Sample	LOI (%)	*t*_1_ (s)	*t*_2_ (s)	Dropping	UL-94 Rating
Pure wood	20.5	>60	-	Yes	NR
SUFW-0	27.2	10.7	14.5	No	V-1
SUFW-1	29.1	5.4	11.8	No	V-0
SUFW-1.5	30.4	3.8	5.6	No	V-0
SUFW-2	32.1	1.2	4.3	No	V-0

**Table 4 polymers-16-01557-t004:** Comparison of flame-retardant properties of flame-retardant materials in this work with those in literature.

Samples	UL-94 Rating	LOI (%)	Reference
SUFW-2	V-0	32.1	This work
P-E/DDM+W	V-0	32.1	[9]
HCPVC-EP	V-0	30.7	[5]
TPLA/PA-HDA_10_	V-0	30	[50]
EP/14IFR/2Ba	V-0	30.5	[22]

**Table 5 polymers-16-01557-t005:** Cone calorimetry data for pure and SUF-coated wood samples (heat flux: 25 kW/m^2^). Peak heat release rate is denoted PHRR, average effective heat of combustion is denoted AV-EHC, and fire growth index is denoted FGI.

Sample	Pure Wood	SUFW-0	SUFW-1	SUFW-1.5	SUFW-2
TTI (s)	95	102	107	173	174
PHRR (kW/m^2^)	246.48	65.56	43.26	13.87	11.25
THR (MJ/m^2^)	12.48	3.50	1.52	1.38	1.06
TSP (m^2^)	52.89	42.88	39.89	31.30	22.32
AV-EHC (MJ/kg)	12.57	7.93	3.15	2.95	2.38
FGI (kW/m^2^/s)	2.24	0.68	0.42	0.08	0.06
Mass (%)	29.72	71.72	79.30	78.43	80.93

**Table 6 polymers-16-01557-t006:** Physical properties of the coatings.

Sample	Pencil Hardness	Adhesion	60° Gloss (Gloss Units)	Water Resistance
SUFW-0	9H	5B	42.3	No effect on surface
SUFW-1	8H	5B	40.4	No effect on surface
SUFW-1.5	8H	5B	35.2	Slight peeling of coating
SUFW-2	8H	5B	31.5	Slight peeling of coating

## Data Availability

The original contributions presented in the study are included in the article, further inquiries can be directed to the corresponding authors.

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
