# Peer review of "Preparation of a Flame-Retardant Curing Agent Based on Phytic Acid–Melamine Ion Crosslinking and Its Application in Wood Coatings"

_polymers, 2024, doi:10.3390/polym16111557_

Round 1

Reviewer 1 Report

Comments and Suggestions for Authors

Review report for polymers-3029780

This manuscript focuses on the ion-crosslinking effect of PA and MEL to prepare a flame-retardant curing agent, PM. Authors reported that flame-retardant coatings of urea formaldehyde resin modified with MEL and sodium lignosulfonate were provided by PM as a flame-retardant curing agent. They also pointed out that the SUFW-2 coated with the flame-retardant coating not only passed UL 94 V-0 level testing but also had a LOI of 32.1%. Furthermore, the SUFW-2 exhibited 95.44% and 91.51% lower PHRR and THR, respectively, in comparison to pure wood and 57.80% lower TSP, showing good flame retardancy and smoke suppression performance. Authors mentioned that the mechanisms of the coatings are mainly due to the formation of an expanded carbon layer during the combustion process, which protects the wooden substrate and reduces smoke generation. Authors added that the physical properties of coatings cured with PM are similar to those of coatings cured with NH4Cl, a commercial curing agent. Finally, they claimed that their results exhibit a new approach for the application of bio-based flame retardants in flame-retardant coatings.

The subject is very interesting and beneficial for the applied purposes. The manuscript has been well organized. However, the concerns raised here must be addressed before manuscript acceptance. My comments are below.

1-      Explain and highlight the novelties of this research work with respect to the previously published papers in the manuscript.

2-      Authors wrote that their results exhibit a new approach for the application of bio-based flame retardants in flame-retardant coatings. Please discus and clarify more this subject.

3-      Please insert CAS numbers the used materials.

4-      Are some prepared samples viscoelastic materials? It is necessary to justify on this issue. Describe and discuss the viscoelastic phenomena, in general, in the introduction, or results and discussion sections. For this purpose, authors can read and cite fully the following references [ [1-book], Non-linear rheology of polymer melts: constitutive equations, rheological properties of polymer blends, shear flow, sliding plate rheometers, LAP Lambert Acad. Publ. 2011, [2-paper] https://doi.org/10.1007/s10965-012-9897-2].

5-      Please perform statistical analysis on the related results. 

6-      Please explain the limitations of this study in the manuscript.

7-      Abbreviations section should be added in the manuscript.

Author Response

Response to reviewers

We gratefully thank the editor and all reviewers for their time spend making their constructive remarks and useful suggestions, which has significantly raised the quality of the manuscript and has enable us to improve the manuscript. Each suggested revision and comment, brought forward by the reviewers was accurately incorporated and considered. Below the comments of the reviews are response point by point and the revisions are indicated.

Reviewer #1: This manuscript focuses on the ion-crosslinking effect of PA and MEL to prepare a flame-retardant curing agent, PM. Authors reported that flame-retardant coatings of urea formaldehyde resin modified with MEL and sodium lignosulfonate were provided by PM as a flame-retardant curing agent. They also pointed out that the SUFW-2 coated with the flame-retardant coating not only passed UL 94 V-0 level testing but also had a LOI of 32.1%. Furthermore, the SUFW-2 exhibited 95.44% and 91.51% lower PHRR and THR, respectively, in comparison to pure wood and 57.80% lower TSP, showing good flame retardancy and smoke suppression performance. Authors mentioned that the mechanisms of the coatings are mainly due to the formation of an expanded carbon layer during the combustion process, which protects the wooden substrate and reduces smoke generation. Authors added that the physical properties of coatings cured with PM are similar to those of coatings cured with NH4Cl, a commercial curing agent. Finally, they claimed that their results exhibit a new approach for the application of bio-based flame retardants in flame-retardant coatings. The subject is very interesting and beneficial for the applied purposes. The manuscript has been well organized. However, the concerns raised here must be addressed before manuscript acceptance.

  1. Explain and highlight the novelties of this research work with respect to the previously published papers in the manuscript.

Response: We feel great thanks for your professional review work on our article. We have added descriptions of the novelty of the work in the abstract, introduction, and conclusion, which mainly reads as follows: “In addition, due to the presence of sodium lignosulfonate, the prepared flame-retardant coating can also dye the material and improve its aesthetic effect. The use of PM has demonstrated the excellent flame retardant and curing effect of bio-based ammonium phytate in urea formaldehyde resin coatings.”

  1. Authors wrote that their results exhibit a new approach for the application of bio-based flame retardants in flame-retardant coatings. Please discus and clarify more this subject.

Response: Thanks for your suggestion. We have added to the novelty of the results of this manuscript that biobased ammonium phytate substances can achieve flame retardant curing of urea-formaldehyde resin coatings. In addition, biomass with similar staining properties to sodium lignosulfonate can be added to achieve the effect that the coating can stain the material.

  1. Please insert CAS numbers the used materials.

Response: Thanks for your suggestion. We have added the CAS numbers of the relevant materials in the Materials and Preparation section of the main text.

  1. Are some prepared samples viscoelastic materials? It is necessary to justify on this issue. Describe and discuss the viscoelastic phenomena, in general, in the introduction, or results and discussion sections. For this purpose, authors can read and cite fully the following references [ [1-book], Non-linear rheology of polymer melts: constitutive equations, rheological properties of polymer blends, shear flow, sliding plate rheometers, LAP Lambert Acad. Publ. 2011, [2-paper] https://doi.org/10.1007/s10965-012-9897-2].

Response: Thanks for your suggestion. We have discussed the viscoelastic phenomenon of materials in the section on the mechanical properties of coatings and cited the relevant content on the strong affinity between epoxy resin and clay surface in the literature "Molecular dynamics study of epoxy/clay nanocomposites: rheology and molecular confinement". The specific reason why coatings can have high adhesion is that hydrogen bonds may form between coatings and wood, and the interaction between coatings and wood increases the strong affinity between coatings and wood surfaces.

  1. Please perform statistical analysis on the related results. 

Response: Thanks for your question. We have statistically analyzed the work with other literature in the Results and discussion section and pointed out the advantages of this work.

  1. Please explain the limitations of this study in the manuscript.

Response: Thanks for your suggestion. We have pointed out the limitations of the study in the conclusion, specifically, the increase of flame-retardant curing agent PM reduces the water resistance of the coating, which needs to be further improved.

  1. Abbreviations section should be added in the manuscript.

Response: Thanks for your careful review. We have supplemented the drug abbreviations in the sample preparation section of the manuscript. The abbreviations for each coating in this work definition are given in Table 1.

Reviewer 2 Report

Comments and Suggestions for Authors

The manuscript by Lixian Sun and coworkers is exceptionally well written and presents the research in a clear and coherent manner. The experimental procedures are described in thorough detail, ensuring that the study can be replicated and understood by readers.

The innovative method of using phytic acid melamine polyelectrolyte (PM) as a flame-retardant agent is particularly noteworthy. By avoiding the use of halides, the authors have demonstrated that it is possible to achieve similar flame-retardant properties. The bio-based approach, leveraging phosphorus-rich biomass and nitrogen-rich melamine, represents a significant advancement in the field of flame-retardant coatings for wood.

The results are compelling, showing a substantial reduction in total heat release, total smoke production, and fire growth index for PM-cured woods compared to uncoated woods. This demonstrates the efficacy of the PM-containing coatings in enhancing flame retardancy and smoke suppression. The mechanical properties of the coatings are also comparable to those cured with commercial curing agents, which underscores the practical applicability of this method.

Overall, this manuscript provides a valuable contribution to the field of bio-based flame retardants and wood coatings. I support the acceptance of this manuscript in Polymers without any changes.

Author Response

Response to reviewers

We gratefully thank the editor and all reviewers for their time spend making their constructive remarks and useful suggestions, which has significantly raised the quality of the manuscript and has enable us to improve the manuscript. Each suggested revision and comment, brought forward by the reviewers was accurately incorporated and considered. Below the comments of the reviews are response point by point and the revisions are indicated.

Reviewer #2: The experimental procedures are described in thorough detail, ensuring that the study can be replicated and understood by readers. The innovative method of using phytic acid melamine polyelectrolyte (PM) as a flame-retardant agent is particularly noteworthy. By avoiding the use of halides, the authors have demonstrated that it is possible to achieve similar flame-retardant properties. The bio-based approach, leveraging phosphorus-rich biomass and nitrogen-rich melamine, represents a significant advancement in the field of flame-retardant coatings for wood.

The results are compelling, showing a substantial reduction in total heat release, total smoke production, and fire growth index for PM-cured woods compared to uncoated woods. This demonstrates the efficacy of the PM-containing coatings in enhancing flame retardancy and smoke suppression. The mechanical properties of the coatings are also comparable to those cured with commercial curing agents, which underscores the practical applicability of this method. Overall, this manuscript provides a valuable contribution to the field of bio-based flame retardants and wood coatings. I support the acceptance of this manuscript in Polymers without any changes.

Response: Thank you very much for your recognition of our work, we will continue to work hard in the future to do better.

Reviewer 3 Report

Comments and Suggestions for Authors

A brief summary: This study presents a new approach for the use of bio-based flame retardants in wood coatings based on phytic acid-melamine ion cross-linking.

Scope: The work is fit with the scope of the journal.

The structure of the paper: It is suggested to create a new section called "Related Works" and put the related works done regarding the fire resistance of wood in this section, and move the related works mentioned in the "Introduction" section to this section. Also, at the end of this section, the existing works should be compared and reviewed, and the disadvantages and limitations that they had, which this research intends to solve, should be stated.

Self-citations: A self-referencing case was observed in reference number 39: Gao, S.; Cheng, Z.; Zhou, X.; Liu, Y.; Chen, R.; Wang, J.; Wang, C.; Chu, F.; Xu, F.; Zhang, D. Unexpected role of amphiphilic lignosulfonate to improve the storage stability of urea formaldehyde resin and its application as adhesives. Int. J. Biol. Macromol. 2020, 161, 755-762, doi:10.1016/j.ijbiomac.2020.06.135.

Up-to-date references: Most of the used references are up-to-date and related to the last 5 years.

Experimental design: The experimental design is suitable for hypothesis testing. It is only necessary to compare the work of the paper with more existing works. For example, resistant and flame-resistant poly (lactic acid) composites produced through reactive compounding with biological ammonium phytate and polyurethane have also been used. reference: " ". Therefore, it is suggested to compare your work with more existing works and examine the advantages of the work expressed in the article compared to the existing works. For example, Tough and flame-retardant poly (lactic acid) composites prepared via reactive blending with biobased ammonium phytate and in situ formed crosslinked polyurethane, have also been done. reference: " Li DF, Zhao X, Jia YW, Wang XL, Wang YZ. Tough and flame-retardant poly (lactic acid) composites prepared via reactive blending with biobased ammonium phytate and in situ formed crosslinked polyurethane. Composites Communications. 2018 Jun 1;8:52-7.". Therefore, it is suggested to compare your work with more existing works and examine the advantages of the work expressed in the article compared to the existing works.

Reproducibility of results: manuscript’s results are reproducible based on the details given in the methods section

Suitability of figures and tables: The figures and tables used in the article are appropriate and related to the stated method. Also, they are easy to interpret and understand

Conclusions section: It is suggested that in the "Conclusion" section, you should briefly compare the work described in the paper with the existing works and express the preference of your work over the existing works. Also, if there are any limitations in this study, mention them.

English Level: The English language is appropriate and understandable.

Comments on the Quality of English Language

The English language is appropriate and understandable.

Author Response

Response to reviewers

We gratefully thank the editor and all reviewers for their time spend making their constructive remarks and useful suggestions, which has significantly raised the quality of the manuscript and has enable us to improve the manuscript. Each suggested revision and comment, brought forward by the reviewers was accurately incorporated and considered. Below the comments of the reviews are response point by point and the revisions are indicated.

Reviewer #3:A brief summary: This study presents a new approach for the use of bio-based flame retardants in wood coatings based on phytic acid-melamine ion cross-linking.

Response: Thanks for your careful review., we will continue to work hard in the future to do better.

Scope: The work is fit with the scope of the journal.

Response: Thank you very much for your recognition of our work, we will continue to work hard in the future to do better.

The structure of the paper: It is suggested to create a new section called "Related Works" and put the related works done regarding the fire resistance of wood in this section, and move the related works mentioned in the "Introduction" section to this section. Also, at the end of this section, the existing works should be compared and reviewed, and the disadvantages and limitations that they had, which this research intends to solve, should be stated.

Response: Thank you for your suggestion. We have added a section on the comparison of our work with other literature in the Results and Discussion section, and the specific results have been presented in the newly added Table 4. At the same time, we pointed out the advantages that this work has compared to other works, which are the use of fewer flame retardants to achieve better flame-retardant effects.

Self-citations: A self-referencing case was observed in reference number 39: Gao, S.; Cheng, Z.; Zhou, X.; Liu, Y.; Chen, R.; Wang, J.; Wang, C.; Chu, F.; Xu, F.; Zhang, D. Unexpected role of amphiphilic lignosulfonate to improve the storage stability of urea formaldehyde resin and its application as adhesives. Int. J. Biol. Macromol. 2020, 161, 755-762, doi:10.1016/j.ijbiomac.2020.06.135.

Response: Thanks for your careful review. After our examination, this literature does not involve self-citation because it does not have the same author as our work.

Up-to-date references: Most of the used references are up-to-date and related to the last 5 years.

Response: Thank you very much for your recognition of our work, we will continue to work hard in the future to do better.

Experimental design: The experimental design is suitable for hypothesis testing. It is only necessary to compare the work of the paper with more existing works. For example, resistant and flame-resistant poly (lactic acid) composites produced through reactive compounding with biological ammonium phytate and polyurethane have also been used. reference: " ". Therefore, it is suggested to compare your work with more existing works and examine the advantages of the work expressed in the article compared to the existing works. For example, Tough and flame-retardant poly (lactic acid) composites prepared via reactive blending with biobased ammonium phytate and in situ formed crosslinked polyurethane, have also been done. reference: " Li DF, Zhao X, Jia YW, Wang XL, Wang YZ. Tough and flame-retardant poly (lactic acid) composites prepared via reactive blending with biobased ammonium phytate and in situ formed crosslinked polyurethane. Composites Communications. 2018 Jun 1;8:52-7.". Therefore, it is suggested to compare your work with more existing works and examine the advantages of the work expressed in the article compared to the existing works.

Response: Thanks for your careful review. We have added a comparison of our work with other literature in the Results and Discussion section. For example, by comparing our work with the literature "Tough and flame-retardant poly(lactic acid) composites prepared via reactive blending with biobased ammonium phytate and in situ formed crosslinked polyurethane", it was found that the flame-retardant additive used in the TPLA/PA-HDA10 prepared in this literature is as high as 10%, while our SUWF-2 flame retardant curing agent is only 2%. Therefore, our work can achieve higher flame-retardant effects at lower flame-retardant content.

Reproducibility of results: manuscript’s results are reproducible based on the details given in the methods section.

Response: Thank you very much for your recognition of our work, we will continue to work hard in the future to do better.

Suitability of figures and tables: The figures and tables used in the article are appropriate and related to the stated method. Also, they are easy to interpret and understand.

Response: Thank you very much for your recognition of our work, we will continue to work hard in the future to do better.

Conclusions section: It is suggested that in the "Conclusion" section, you should briefly compare the work described in the paper with the existing works and express the preference of your work over the existing works. Also, if there are any limitations in this study, mention them.

Response: Thanks for your suggestion. We have added a comparison with existing works in the conclusion section and expressed the advantages of the work compared to existing works. Specifically, the flame-retardant curing agent prepared in the work has advantages such as simple preparation, low dosage, and excellent improvement of flame-retardant performance compared to existing works. At the same time, it was mentioned that the limitation of this work is that an increase in the amount of PM prepared in this work will lead to a decrease in the water resistance performance of the coating, which requires further improvement in the work.

English Level: The English language is appropriate and understandable.

Response: Thank you very much for your recognition of our work, we will continue to work hard in the future to do better.

Round 2

Reviewer 1 Report

Comments and Suggestions for Authors

The authors have adequately addressed the reviewer comments. The revised manuscript is now recommended for publication at the present form.